



# Measurement Report: Optical properties and sources of water-soluble brown carbon in Tianjin, North China: insights from organic molecular compositions

Junjun Deng[1], Hao Ma[1], Xinfeng Wang[2], Shujun Zhong[1], Zhimin Zhang[1], Jialei Zhu[1], Yanbing Fan[1],
Wei Hu[1], Libin Wu[1], Xiaodong Li[1], Lujie Ren[1], Chandra Mouli Pavuluri[1], Xiaole Pan[3], Yele Sun[3], Zifa
Wang[3], Kimitaka Kawamura[4], and Pingqing Fu[1]

[1]Institute of Surface-Earth System Science, School of Earth System Science, Tianjin University, Tianjin 300072, China
[2]Environment Research Institute, Shandong University, Jinan 250100, China
[3]State Key Laboratory of Atmospheric Boundary Layer Physics and Atmospheric Chemistry, Institute of Atmospheric
Physics, Chinese Academy of Sciences, Beijing 100029, China
[4]Chubu Institute for Advanced Studies, Chubu University, Kasugai 487-8501, Japan

*Correspondence to:* Pingqing Fu (fupingqing@tju.edu.cn)

**Abstract.** Brown carbon (BrC) aerosols exert vital impacts on climate change and atmospheric photochemistry due to their light absorption in the wavelength range from near-ultraviolet (UV) to visible light. However, the optical properties and
formation mechanisms of ambient BrC remain poorly understood, limiting the estimation of their radiative forcing. In the present study, fine aerosols ($PM_{2.5}$) were collected during 2016–2017 on a day/night basis over urban Tianjin, a megacity in North China, to obtain seasonal and diurnal patterns of atmospheric water-soluble BrC. There were obvious seasonal but no evident diurnal variations in light absorption properties of BrC. In winter, BrC showed much stronger light absorbing ability since mass absorption efficiency at 365 nm ($MAE_{365}$) ($1.54 \pm 0.33$ $m^2$ $g^{-1}$), which was 1.8 times larger than that ($0.84 \pm 0.22$
$m^2$ $g^{-1}$) in summer. Direct radiative effects by BrC absorption relative to black carbon in the UV range were $54.3 \pm 16.9$ % and $44.6 \pm 13.9$ %, respectively. In addition, five fluorescent components in BrC, including three humic-like fluorophores and two protein-like fluorophores were identified with excitation-emission matrix fluorescence spectrometry and parallel factor (PARAFAC) analysis. The lowly-oxygenated components contributed more to winter and nighttime samples, while more-oxygenated components increased in summer and daytime samples. The higher humification index (HIX) together with lower
biological index (BIX) and fluorescence index (FI) suggest that the chemical compositions of BrC were associated with a high aromaticity degree in summer and daytime due to photobleaching. Fluorescent properties indicate that wintertime BrC were predominantly affected by primary emissions and fresh secondary organic aerosol (SOA), while summer ones were more influenced by aging processes. Results of source apportionments using organic molecular compositions of the same set of aerosols reveal that fossil fuel combustion and aging processes, primary bioaerosol emission, biomass burning, and biogenic
and anthropogenic SOA formation were the main sources of BrC. Biomass burning contributed much larger to BrC in winter and at nighttime, while biogenic SOA contributed more in summer and at daytime. Especially, our study highlights that primary bioaerosol emission is an important source of BrC in urban Tianjin in summer.



# 1 Introduction

Brown carbon (BrC) is light absorbing organic carbon (OC) in the atmosphere, which can absorb radiation in the range from
near-ultraviolet (UV) to visible and show strong wavelength dependence (Andreae and Gelencsér, 2006; Bahadur et al., 2012).
Although its light absorbing ability is generally weaker than that of black carbon (BC), BrC exerts considerable impacts on
atmospheric radiative balance and global climate due to their large abundance and strong light absorption in the near-UV
spectrum (Feng et al., 2013; Jo et al., 2016; Zhang et al., 2017; Zhang et al., 2020a). In addition, BrC can efficiently affect the
atmospheric photochemistry processes, formation of secondary organic aerosol (SOA) and thus regional air quality by
influencing the photolysis rates of atmospheric radicals (Laskin et al., 2015; Moise et al., 2015; Mok et al., 2016; Baylon et
al., 2018). Since the last decade, plenty of studies on BrC aerosols have been performed to explore their optical properties and
estimate their environmental and climatic effects (Hecobian et al., 2010; Kirillova et al., 2016; Liu et al., 2016; Huang et al.,
2018; Shamjad et al., 2018; Li et al., 2020c; Choudhary et al., 2021).

However, it is quite a challenge to understand the extremely complex chemical composition, sources, and formation and
evolution mechanisms of BrC (Laskin et al., 2015; Yan et al., 2018; Wu et al., 2021). On the one hand, atmospheric BrC is
derived from incomplete combustion of carbonaceous materials, such as fossil fuel, biomass and biofuel (Chakrabarty et al.,
2010; Lack et al., 2012; Lin et al., 2016, 2017; Sun et al., 2017; Lei et al., 2018; Hettiyadura et al., 2021). One the other hand,
BrC can be formed through aqueous-phase or heterogeneous reactions from both biogenic and anthropogenic precursors (Lin
et al., 2015; Li et al., 2020a; He et al., 2021). Furthermore, optical properties and chemical compositions of BrC aerosols will
undergo significant changes when they are experiencing atmospheric physical and chemical processes such as photochemical
aging and hygroscopic growth (Lee et al., 2014; Forrister et al., 2015; Sumlin et al., 2017; Wong et al., 2017; Dasari et al.,
2019; Kasthuriarachchi et al., 2020; Palm et al., 2020; Ni et al., 2021). Therefore, despite the progresses reported in recent
years, it is needed to further characterize the sources and formation mechanisms of atmospheric BrC, particularly from the
perspectives of chromophore and molecular composition (Laskin et al., 2015; Yan et al., 2018).

To identify the chromophores in BrC will be benefit for probing the sources, dynamic optical properties, and aging processes
of atmospheric BrC (Laskin et al., 2015; Yan et al., 2020). However, to our best knowledge, until now it is still challengeable
to conduct a comprehensive analysis of the chromophores of atmospheric BrC. One great difficulty is to distinguish the
absorbing chromophores from a majority of nonchromophoric components. Three-dimensional excitation-emission matrix
(EEM) fluorescence spectroscopy is a powerful tool for revealing the chemical compositions, sources, and chemical reactions
of complex chromophores in different environments since each chromophore has its own specific excitation-emission peak in
the EEM maps (Coble, 1996, 2007; Murphy et al., 2013). In recent years, the fluorescence technique has been used to
investigate the characteristics and potential sources of chromophores in aerosols (Mladenov et al., 2011; Fu et al., 2015; Chen
et al., 2016, 2020; Qin et al., 2018; Wang et al., 2020; Tang et al., 2020). With the fluorescence technique, some categories of
chromophores such as humic-like chromophores and protein-like chromophores can be identified in atmospheric aerosols.
However, applications of the fluorescence technique in atmospheric BrC aerosols were comparatively limited so far (Wu et





al., 2021).

The North China Plain (NCP), with a regional population contribution of approximately 25%, is the second largest plain in China. The NCP is also one of the most developed city clusters in China and contains several megacities such as Beijing, Tianjin, and Shijiazhuang. Due to the rapid economic development and intensive anthropogenic activities, the NCP has been

suffering severe regional air pollution in the recent years, which has attracted a world-wide concern (Zhao et al., 2013; Guo et al., 2014; Huang et al., 2017; Gao et al., 2018; Ge et al., 2018; Li et al., 2021; Zhang et al., 2021). Despite numerous studies on chemical compositions, source apportionment and formation mechanisms of atmospheric aerosols, current understanding of the optical properties and sources of BrC aerosols over the NCP are still inadequate. Nevertheless, the limited researches focusing on BrC aerosols are mostly conducted in Beijing (Cheng et al., 2011; Du et al., 2014; Yan et al., 2015, 2020; Li et

al., 2020e), while the BrC-related studies in other cities in this hot-spot region are quite scarce and therefore deserve more attention. Located adjacent to Beijing and the Bohai Sea, Tianjin is the largest industrial city and second largest megacity in North China. Previous studies have found that Tianjin experienced serious aerosol pollution with large contribution from anthropogenic activities including coal combustion, industrial and vehicle emissions (Huang et al., 2017; Gao et al., 2018). Abundances and molecular compositions of organic aerosols were investigated and the contributions of primary emission

sources and secondary formation to organic aerosols were also evaluated (Fan et al., 2020b). However, optical properties and formation mechanisms of BrC in Tianjin are still unclear.

In the present study, field measurements of water-soluble BrC in ambient fine aerosols were performed in urban Tianjin. Seasonal and diurnal variations in optical properties of BrC were investigated, and direct radiative effects by BrC aerosols were also estimated. The fluorescence technique was adopted to further explore the components and possible chromophores

in BrC. The impacts of various sources and photooxidation on atmospheric BrC were unveiled by analyzing the relationships of BrC with chemical compositions and organic molecular markers of aerosols. This study provides a comprehensive view on the temporal variability in optical properties and sources of BrC, helping to deepen the understanding in its climatic effects.

## 2 Methods

### 2.1 Sample Collection

$PM_{2.5}$ (Particulate matters with aerodynamic diameter < 2.5 μm) sampling was performed using a high-volume sampler (Tisch TE-PM$_{2.5}$ HVP-BL) at a flow rate of 1.05 m$^3$ min$^{-1}$. The air sampler was equipped on the rooftop (~20 m above ground level) of a building on the Weijinlu Campus of Tianjin University (39.11º N, 117.17º E) in urban Tianjin. The sampling site is close to commercial and residential region, and there is no obvious industrial emission around the site. Field campaigns were conducted from 10 November to 23 December 2016 (winter) and from 22 May to 22 June 2017 (summer). Daytime samples

were collected from 08:00 to 20:00 LT, and nighttime samples were collected from 20:00 to 08:00 LT next day. Aerosols were collected onto quartz fiber filters with a size of 8×10 inch (Pallflex 2500QAT-UP), which were preheated at 450 ℃ for 6 h in a muffle furnace to remove potential contamination from organics. Field blank filters were also collected during both seasons



by keeping blank filter in the sampler for 5 min without air flow. After collection, the samples were stored in the dark at – 20 °C until analysis.

## 2.2 Chemical Analysis

Two pieces of quartz filters with diameter of 14 mm were punched and extracted with 25 mL ultrapure water (> 18.2 MΩ cm). The extracts were under ultrasonication for 20 min and then filtered through a poly tetra fluoroethylene (PTFE) syringe filter (0.22 μm) to remove water-insoluble compounds. Concentrations of water-soluble organic carbon (WSOC) in the water extracts were determined with a TOC analyzer (TOC-VCPH, Shimadzu, Japan). The rest extracts were used for the light absorption and fluorescence measurement. A punch of quartz filter with an area of ~2.3 cm$^2$ was cut to measure the EC and OC concentrations in aerosol samples by a thermal-optical carbon analyzer (Model RT-4, Sunset, USA), following the National Institute for Occupational Safety and Health (NIOSH) protocol. Secondary organic carbon (SOC) was estimated with the EC tracer method (Castro et al., 1999). Major water-soluble inorganic ions (e.g., $SO_4^{2-}$, $NO_3^-$, $Cl^-$, $NH_4^+$, and $K^+$) were analyzed with ion chromatography (ICS 5000+, Thermo, USA).

The measurements of molecular markers in organic aerosols have been reported in detail in our previous study (Fan et al., 2020b) and are briefly described here. First, a filter aliquot was extracted with dichloromethane/methanol (2:1; v/v) under ultrasonication for 10 min for three times. The extracts were then concentrated with a rotary evaporator, and dried with pure nitrogen gas. After that, 50 μL of N,O-*bis*-(trimethylsilyl)trifluoroacetamide (BSTFA) with 1 % trimethylsilyl chloride and 10 μL of pyridine was added to the extracts to react at 70 °C for 3 h. After the polar groups were derivatized into the trimethylsilyl (TMS) easters and ethers, the derivatives were added to 40 μL of *n*-hexane-containing internal standards (C$_{13}$ *n*-alkane, 1.43 ng μL$^{-1}$) before gas chromatography/mass spectrometry (GC/MS) analysis. GC/MS analysis was performed using an Agilent model 7890 GC coupled to a 5975c mass-selective detector to identify and quantify organic compound classes. The GC was equipped with a split/splitless injector and a DB-5MS fused silica capillary column (30 m × 0.25 mm i.d. 0.5 μm film thickness). The samples in the fused silica capillary column were analyzed using a specific GC temperature program with the carrier gas of helium. The MS was operated on electron impact (EI) mode at 70 eV, scanning from 50 to 650 Da. Data processes were performed with the Chemstation software. The results were corrected by field blanks, which were treated as the real samples. Temporal variations in concentrations of carbonaceous species and some molecular markers in PM$_{2.5}$ including sugars and SOA tracers are plotted in Figure S1, S2, and S3, and seasonal average concentrations are summarized in Table S1.

## 2.3 Light Absorption Analysis

Light absorbance (A$_\lambda$) of the water extracts at the wavelength (λ) spectra between 200 and 700 nm was measured with a UV-Vis spectrophotometer (UV2700, Shimadzu). Light absorption coefficient Abs$_\lambda$ (Mm$^{-1}$) of the dissolved organic matter (DOM) at the wavelength λ can be calculated as follows:



$$Abs_\lambda = (A_\lambda - A_{700}) \times \frac{V_l}{V_a \times L} \times \ln(10), \tag{1}$$

where $V_l$ is the volume of the extracts, $V_a$ is the volume of the punched and extracted air, and $L$ is the optical path length (0.01
m in this study). $A_\lambda$ are referenced to the $A_{700}$ to account for any baseline drift (Hecobian et al., 2010). In this study, light
absorption coefficients of water-soluble organics at 365 nm ($Abs_{365}$) are used as proxy of BrC in accordance with previous
studies (Laskin et al., 2015).

The mass absorption efficiency (MAE: $m^2\ g^{-1}$) of water-soluble BrC can be derived as follows:

$$MAE_\lambda = \frac{Abs_\lambda}{[WSOC]}, \tag{2}$$

where $[WSOC]$ ($\mu gC\ m^{-3}$) represents the mass concentration of WSOC.

The wavelength dependence of light absorption fits a power law as follows:

$$Abs_\lambda = C \times \lambda^{-AAE}, \tag{3}$$

where $C$ is a concentration- and composition-related constant, and AAE is the absorption Ångström exponent depending on
the types of chromophores. In this study, AAE was fitted at the range of 300–450 nm.

The particle refractive index ($m = n + k\mathrm{i}$) is a key optical parameter in climate model expressing the light extinction ability of
ambient aerosols. The imaginary part $k$ represents light absorption and can be estimated from MAE as follows (Liu et al.,
2013):

$$k_\lambda = \frac{\lambda \times \rho \times Abs_\lambda}{4\pi \times [WSOC]} = \frac{\lambda \times \rho \times MAE_\lambda}{4\pi}, \tag{4}$$

where $\rho$ ($g\ m^{-3}$) is particle density and assumed as 1.5 according to Liu et al. (2013).

**2.4 Determination of Direct Radiative Absorption by BrC**

The direct radiative forcing of BrC in Tianjin was assessed with the simple forcing efficiency (SFE). SFE ($W\ g^{-1}$) represents
the energy added to the Earth-atmosphere system by per unit mass aerosol (Bond and Bergstrom, 2006). The wavelength-
dependent SFE of BrC can be calculated as follows (Chen and Bond, 2010):

$$\frac{dSFE}{d\lambda} = -\frac{1}{4} \frac{dS(\lambda)}{d\lambda} \tau_{atm}^2 (\lambda)(1 - F_c) \left[ 2(1 - a_s)^2 \beta(\lambda) MSE(\lambda) - 4a_s MAE(\lambda) \right], \tag{5}$$

where $dS(\lambda)/d\lambda$ is the wavelength-dependent solar irradiance, $\tau_{atm}$ is the atmospheric transmission (0.79), $F_c$ is the cloud
fraction (0.6), $\alpha_s$ is the surface albedo (0.19 for global average), $\beta$ is the backscatter fraction, and MSE and MAE are the mass
scattering and absorption efficiency of BrC, respectively.





Direct radiative forcing due to aerosol scattering can be ignored when estimating the radiative effects of BrC light absorption. Therefore, the absorption radiative forcing in a given spectral range was calculated by integrating the SFE values per nanometer

with the simplified Eq.5 as follows:

$$SFE = \int \frac{dS(\lambda)}{d\lambda}\tau^2(\lambda)(1-F_c)a_s MAE(\lambda)d\lambda,$$    (6)

In addition, the relative direct climate warming effects due to BrC light absorption were also estimated by comparing the direct radiative forcing of BrC with that of BC (Bosch et al., 2014; Kirillova et al., 2014). Relative radiative forcing of BrC ($f_{BrC}$) is calculated with the method as follows:

$$f_{BrC} = \frac{\int I_0(\lambda)\cdot\left\{1-e^{-\left(MAE_{BrC,365}\left(\frac{365}{\lambda}\right)^{AAE}\cdot[BrC]\cdot h_{ABL}\right)}\right\}d\lambda}{\int I_0(\lambda)\cdot\left\{1-e^{-\left(MAE_{BC,550}\left(\frac{550}{\lambda}\right)\cdot[BC]\cdot h_{ABL}\right)}\right\}d\lambda},$$    (7)

where $I_0(\lambda)$ is the solar emission spectrum estimated using the clear sky Air Mass 1 Global Horizontal (AM1GH) irradiance model (Levinson et al., 2010); $MAE_{BC,550}$ is the mass absorption efficiency for BC at 550 nm, which is set to 7.5 m$^2$ g$^{-1}$ and the AAE for BC is set to 1 based on Bond and Bergstrom (2006) and Kirillova et al. (2014); [BC] is the mass concentration of BC; The height of the atmospheric boundary layer ($h_{ABL}$) is adopted as 1000 m because it has little impact on the calculated

ratio in the range of 200–3000 m (Kirillova et al., 2014).

**2.5 Fluorescence Analysis**

The EEM fluorescence spectra of the extracts were measured using a fluorometer (Aqualog, Horiba). The excitation wavelength range was 240–550 nm (3 nm interval), and the emission wavelength range was 246–828 nm (~2.4 nm interval). The measured EEM spectra were calibrated by instrument calibration, internal filter correction and Raman correction (Murphy

et al., 2013). The EEM spectra of all samples were corrected by subtracting the blank sample. The fluorescence intensities were further divided by the amount of water used for the extraction and the air volume of each filter sample to convert the fluorescence unit to RU m$^{-3}$. The fluorescence properties of the extracts were determined through humification index (HIX), biological index (BIX), and fluorescence index (FI). FI was determined by the ratio of emission intensity of 450 to 500 nm under the excitation wavelength of 370 nm, BIX was determined by the ratio of emission intensity of 380 nm to 430 nm under

the excitation wavelength of 310 nm, HIX was determined by the ratio of the integrated fluorescence emission intensity in the range of 435–480 nm to 300–345 nm under the excitation wavelength of 255 nm (Battin, 1998; McKnight et al., 2001). With the fully corrected and treated EEM fluorescence spectra data, the parallel factor analysis (PARAFAC) model was adopted to identify the fluorescent components of BrC (Murphy et al., 2013). PARAFAC is a mathematical method to separate chemically



independent but spectrally overlapping fluorescent components based on assumption that EEM spectra are independent, liner
related, and additive (Murphy et al., 2011). During the recent years, PARAFAC model has been used to investigate The
fluorescence properties of aerosol WSOC (Pöhlker et al., 2012; Matos et al., 2015; Chen et al., 2016, 2020; Wu et al., 2019,
2020b; Dey et al., 2021). The PARAFAC modeling was performed with the software package Solo (Eigenvector Inc.).

**2.6 Source apportionment of BrC**

Positive Matrix Factorization (PMF, version 5.0), a receptor model developed by the United States Environmental Protection
Agency (USEPA), was adopted to carry out the source apportionment of BrC. PMF model is a multivariate factor analysis tool
that decomposes a measured sample matrix into two matrices including factor profiles and factor contributions (Paatero and
Tapper, 1994). PMF can provide as model outcome both the source profiles and contributions of various sources without
inputting source profiles. In this study, the measurement data of BrC (i.e., $Abs_{365}$) and chemical species of the aerosol samples,
including OC, WSOC and major inorganic ions, were selected as inputs of PMF model. Especially, the organic molecular
markers (i.e., sugars, biogenic and anthropogenic SOA compounds) were implemented to the PMF model to constrain the
sources of BrC. Separating and identifying different source factors with molecular markers species enables more accurate and
finer results of source apportionment of organic aerosols (Wang et al., 2017; Al-Naiema et al., 2018; Li et al., 2020d).

**3 Results and Discussion**

**3.1 Light Absorption Properties of BrC**

The variations in light absorption coefficients of water-soluble BrC with wavelength in the spectral range of 300–600 nm in
winter and summer in Tianjin are presented in Figure S4. The absorption spectrums showed an evident feature of BrC, since
they were highly wavelength-dependent and decreased remarkably from the ultraviolet to the visible ranges. BrC light
absorption was more wavelength-dependent in winter than in summer, since the winter average AAE of BrC was 5.4 ± 0.4,
about 10 % higher than the summer average (4.9 ± 0.6) (Table 1). However, AAE in summer varied in a relatively wider range
(3.2–6.5) compared with that in winter (4.3–6.1) (Figure 1a). The gap between AAE values in different seasons indicate the
distinct chemical composition of BrC resulted from various sources and atmospheric formation/aging processes. BrC in winter
may be significantly affected by primary emissions of fossil fuel combustion, since high AAE coefficients are often associated
with biomass burning (Desyaterik et al., 2013) and coal combustion (Li et al., 2019). In contrast to the significant seasonal
variations, there were no evident diurnal variations in AAE, because daytime AAE values (4.8 ± 0.6 in summer and 5.4 ± 0.4
in winter) were comparable with nighttime AAE values (4.9 ± 0.6 in summer and 5.4 ± 0.3 in winter). *t*-test results also showed
that the day/night differences in AAE values were not significant. It indicates that chemical compositions of BrC in daytime
and nighttime may be similar.



Water-soluble BrC light absorption (at 365 nm) (i.e., $Abs_{365}$) in Tianjin experienced obvious day-to-day variations in both winter and summer (Figure 1b). Exhibiting remarkable seasonal variations, $Abs_{365}$ values were much larger in winter than in

summer, similar to the concentrations of WSOC and OC (Table S1). $Abs_{365}$ was in the range of 2.0–53.7 $Mm^{-1}$ in winter and 0.5–6.1 $Mm^{-1}$ in summer. The average $Abs_{365}$ was 14.1 ± 8.5 $Mm^{-1}$ in winter, ~6.7 times higher than the summer average (2.1 ± 1.0 $Mm^{-1}$) (Table 1). Generally, there are no significant differences between daytime and nighttime $Abs_{365}$, with the daytime averages of 14.4 ± 10.3 $Mm^{-1}$ (winter) and 2.0 ± 0.8 $Mm^{-1}$ (summer) and nighttime averages of 13.9 ± 6.3 $Mm^{-1}$ (winter) and 2.1 ± 1.1 $Mm^{-1}$ (summer). Temporal variations in BrC light absorption are closely related to both the abundance and absorption

capacity of BrC.

MAE can be used to describe the light absorbing ability of BrC aerosols. $MAE_{365}$ values of water-soluble BrC in winter and summer were in the ranges of 1.06–2.58 $m^2\ g^{-1}$ and 0.36–1.50 $m^2\ g^{-1}$, respectively (Figure 1c). Average $MAE_{365}$ in winter (1.54 ± 0.33 $m^2\ g^{-1}$) was ~1.8 times higher than that in summer (0.84 ± 0.22 $m^2\ g^{-1}$), suggesting the much higher light absorption capacity of BrC in winter than in summer (Table 1). The imaginary refractive index, $k$, is a vital parameter representing the

light-absorbing ability used in climate model to assess direct radiative forcing of aerosols (Andreae and Gelencsér, 2006; Shamjad et al., 2016). $k_{365}$ for water-soluble BrC in Tianjin were in the range of 0.052–0.127 in winter and 0.018–0.074 in summer, with the seasonal averages of 0.076 ± 0.016 and 0.041 ± 0.011, respectively (Table 1). The obvious seasonal variations in both $MAE_{365}$ and $k_{365}$ also suggest the distinct sources and formation mechanisms of BrC chromophores in different seasons. In contrast, although the nighttime light-absorbing ability was slightly stronger than in daytime in both seasons, diurnal

differences were not statistically significant between daytime and nighttime $MAE_{365}$ (and $k_{365}$) as revealed by $t$-test, also indicating the similar chemical compositions of BrC in daytime and nighttime. $MAE_{365}$ values in Tianjin were comparable with those in Beijing in North China (Yan et al., 2015), higher than those in the US (Xie et al., 2019) and Europe (Moschos et al., 2018), and lower than those in New Delhi and Kanpur in India (Dasari et al., 2019; Choudhary et al., 2021).

### 3.2 Direct Radiative Absorption by BrC

Radiative forcing efficiency of water-soluble BrC was estimated by integrating the wavelength-dependent SFE from 300 to 700 nm (i.e., $SFE_{300–700}$). Because BrC mainly absorb solar radiation in the UV spectral region, the BrC absorption radiative forcing efficiency in the 300−400 nm range (i.e., $SFE_{300–400}$) was also calculated. Figure 1d illustrates the temporal variations in the radiative forcing efficiencies of BrC at the two spectra. In summer, $SFE_{300–400}$ and $SFE_{300–700}$ varied from 0.6 to 2.4 W $g^{-1}$ and 1.7 to 10.5 W $g^{-1}$, respectively. By comparison, variations in the forcing efficiencies were slightly larger in winter, and

$SFE_{300–400}$ and $SFE_{300–700}$ varied from 1.6 to 3.6 W $g^{-1}$ and 3.3 to 13.4 W $g^{-1}$. Average BrC forcing efficiency in winter (summer) were 6.2 ± 2.0 W $g^{-1}$ (4.6 ± 1.7 W $g^{-1}$) over the entire solar spectrum and 2.4 ± 0.5 W $g^{-1}$ (1.4 ± 0.4 W $g^{-1}$) in the UV range (Table 1). $SFE_{300–400}$ and $SFE_{300–700}$ were ~71 % and ~35 % larger in winter than summer, respectively, indicating the more abundant BrC with stronger light-absorbing capacity resulted in a remarkable increase in direct radiative forcing by BrC. It should be noted that $SFE_{300–400}$ accounted for 22.4–57.4 % (40.3 ± 6.4 %) and 21.0–52.0 % (30.7 ± 5.8 %) of $SFE_{300–700}$ in

winter and summer, respectively, suggesting radiative forcing in the UV range plays a vital role in radiative forcing by BrC



absorption. Comparing with the limited literature, the BrC forcing efficiency in Tianjin was slightly larger than that in Hong Kong (4.4 W g$^{-1}$ in winter, Zhang et al., 2020b), while much smaller than that in Xi'an in Northwest China (11.7 W g$^{-1}$ in winter, Zhang et al., 2020b) and Kanpur in India (19.2 W g$^{-1}$ in winter and 12.3 W g$^{-1}$ in monsoon, Choudhary et al., 2021). Direct radiative forcing by BrC absorption was also evaluated by calculating solar radiative effect of water-soluble BrC relative

to BC in the 280−4000 nm range (i.e., $f_{280-4000}$). Relative radiative effect of BrC in the UV spectral region (i.e., $f_{300-400}$) was also calculated. $f_{280-4000}$ was in the ranges of 4.5–25.3 % and 4.8–25.6 % (Figure 1e), with the comparable averages of 13.5 ± 4.1 % and 12.5 ± 4.5 % in winter and summer (Table 1), respectively, indicating that water-soluble BrC is a non-negligible contributor to the climate warming by absorbing solar radiation. Although BrC light absorption was much stronger in winter than summer, relative direct radiative effects of BrC at the entire solar spectrum were comparable in the two seasons. It can be

attributed to the enhanced direct radiative effect by BC absorption due to the sharp increase in BC concentration in winter (Table S1). Relative direct radiative effect in the UV range ($f_{300-400}$) were in the ranges of 19.0–96.1 % and 17.4–76.6 % in winter and summer, respectively. $f_{300-400}$ exhibited obvious seasonal variations with a much larger value in winter (54.3 ± 16.9 %) than in summer (44.6 ± 13.9 %). The much larger $f_{300-400}$ compared with $f_{280-4000}$ suggested that although BC still dominated radiative effect by light-absorbing carbonaceous aerosols, BrC played a far more important role in the shorter

wavelength in comparison to the entire spectrum. The large contributions of BrC light absorption in the UV spectral range deserve more attentions due to its potential impact on atmospheric photochemistry and ozone formation (Mok et al., 2016; Baylon et al., 2018). Relative direct radiative effects of water-soluble BrC in Tianjin were comparable to those in High Arctic (13 ± 7 %) (Yue et al., 2019a), and much larger than those in other urban locations in China such as Beijing (5.7 ± 2.5 % in summer and 10.7 ± 3.0 % in winter) (Yan et al., 2015), and Xi'an (2 ± 1 % in summer and 10 ± 4 % in winter) (Huang et al.,

260 2018).

### 3.3 Fluorescence Indices of BrC

Fluorescence indices originally developed as indicators of the type and source of the fluorescent DOM in aquatic systems and soils have been applied to investigate the sources and aging processes of organic aerosols for a decade (Mladenov et al., 2011; Lee et al., 2013; Fu et al., 2015; Qin et al., 2018; Tang et al., 2021; Wu et al., 2021). HIX is a proxy for the aromaticity of

DOM and an increased HIX value is usually accompanied a higher polycondensation degree, C/H ratio, and aromaticity of DOM (Zsolnay et al., 1999; McKnight et al., 2001; Birdwell and Engel, 2010). HIX values in this study were 2.22 ± 0.54 (1.17–3.51) and 2.73 ± 0.51 (1.64–3.96) during winter and summer, respectively (Figure 1f and Table 1), suggesting that water-soluble organic aerosols were less aromatic compared with aquatic or soil DOM, which might be attributed to the lower molecular weight and smaller contributions from aromatic organics (Qin et al., 2018). HIX values of Tianjin aerosols were

comparable to those of aerosols in Mt. Tai, North China (1.7–3.4, 2.4) (Yue et al., 2019b) and the Colorado Rocky Mountains, USA (0.72–4.75, 2.42) (Xie et al., 2016), lower than those in Indo-Gangetic Plain, India (4.8 ± 0.3) (Dey et al., 2021), Bangkok, Thailand (3.4 ± 0.99) (Tang et al., 2021) and the high Arctic (0.69–5.24, 2.93) (Fu et al., 2015), and higher than those in Lanzhou, China (1.2 in winter and 2.0 in summer) (Qin et al., 2018), suggesting the moderate aromaticity degree of water-





soluble BrC in Tianjin. Similar to previous research in Lanzhou, Northwest China (Qin et al., 2018), HIX values in Tianjin

experienced obvious seasonal variations with higher values in summer than winter, suggesting that water-soluble BrC in summer had a higher aromaticity degree or increasing polycondensation in chemical structure (Zsolnay et al., 1999). Low HIX values were probably associated with freshly introduced primary organic aerosols and fresh secondary organic aerosols (SOA); however, HIX values would significantly increase during the aging processes of organic aerosols (Lee et al., 2013; Tang et al., 2021). Therefore, HIX values indicated that BrC in Tianjin were significantly affected by primary emissions and less aged in

winter, while more aged in summer due to the strong photooxidation and secondary chemical processes.

FI and BIX were both adopted to assess the relative contributions from biological sources to DOM. The fluorophore is often associated with higher aromaticity if FI is low, and vice versa (McKnight et al., 2001; Fu et al., 2015). High BIX usually corresponds to the predominant biological or microbial materials, while low BIX indicates few biological organics (Huguet et al., 2009). For water-soluble BrC in Tianjin, BIX values were $1.32 \pm 0.14$ (1.08–1.73) and $1.19 \pm 0.13$ (0.92–1.45) in winter

and summer, and the corresponding FI values were $1.71 \pm 0.06$ (1.60–1.88) and $1.61 \pm 0.10$ (1.32–1.82), respectively (Figure 1g–h and Table 1). The slightly lower summer values of BIX and FI indicate that fluorescent BrC aerosols in summer had higher aromaticity degrees, in accordance with the HIX results. The lower FI values in summer may be a result of photobleaching of fluorescent DOM, since fluorescent DOM absorbing light at higher wavelengths would be removed due to photochemical processes (McKnight et al., 2001; Xie et al., 2016). Another possible reason for the higher BIX and FI values

in winter was that primary aerosols from coal combustion and biomass burning had high FI and BIX values (Tang et al., 2021). Table 1 also shows that in both seasons BIX and FI values were slightly lower at daytime compared with nighttime, suggesting that fluorophores in aerosols at daytime were more aged due to photobleaching and thus had higher aromaticity.

Figure 1 presents an obvious seesaw relationship between HIX and BIX (or FI) in winter. HIX showed significantly negative correlations with both FI ($R = -0.803$, $p < 0.01$) (Figure 2a) and BIX ($R = -0.927$, $p < 0.01$) (Figure 2b) in winter, indicating

the quite similar factors controlling aromaticity and biological contribution of organic aerosols. Actually, the elevated aromaticity in winter were mainly led by the increased aromatic compounds (e.g., polycyclic aromatic hydrocarbons (PAHs)), which were mostly emitted from anthropogenic sources. Therefore, higher aromaticity degrees were generally associated with larger contribution from anthropogenic sources while smaller contribution from biological activities. However, in summer, correlations of HIX with FI ($R = -0.207$, $p > 0.05$) or and ($R = -0.130$, $p > 0.05$) were not significant, suggesting the different

influencing factors of aromaticity and biological contribution during the period. For example, secondary formation and aging processes play an important role in the hot season due to the stronger radiation and higher temperature, which may result in increases in both HIX and BIX, although the influences on BIX are relatively small (Lee et al., 2013). Comparison of HIX as a function of FI and BIX for Tianjin aerosols together with aerosol samples in literature are summarized in Figure 2c and 2d, respectively. HIX values for Tianjin aerosols mainly concentrated in the region where freshly emitted aerosols (Mladenov et

al., 2011) and fresh SOA (Lee et al., 2013) located, but they were much lower than HIX values for aged SOA (Lee et al., 2013) and aged dust aerosols experiencing long-range transport (Mladenov et al., 2011). BIX and FI values of aerosols in Tianjin were mainly located within the region where the primary aerosols from biomass burning, coal combustion, and vehicle



emissions concentrated (Tang et al., 2021). Therefore, the fluorescence indices together indicated water-soluble BrC in Tianjin prominently contained freshly emitted and less aged aerosol, especially in winter. However, since the influencing mechanisms

of fluorescence properties of atmospheric organic compounds are extremely complicated and still unclear, further studies on fluorescence indices are needed (Wu et al., 2021).

**3.4 Fluorescent Components of BrC**

To explore the possible sources and controlling factors of fluorescent BrC, correlations of fluorescent intensities with light absorption and chemical compositions of aerosols were examined. Generally, fluorescent intensities were strongly correlated

with $Abs_{365}$ in both winter ($R = 0.863$, $p < 0.01$) and summer ($R = 0.882$, $p < 0.01$), suggesting that the affecting mechanisms of light absorption and fluorescent properties were much similar and a majority of light-absorbing BrC aerosols were fluorescent (Figure S5a). The strong correlations of fluorescent intensities with EC ($R = 0.743$–$0.841$, $p < 0.01$) (Figure S5b) and SOC ($R = 0.484$–$0.820$, $p < 0.01$) (Figure S5c) suggested the combined effects of combustion-related sources and secondary formation processes on BrC fluorophores. Note that fluorescent intensities even showed a stronger correlation with

SOC than EC in winter, indicating the dominant source of fluorescent BrC might be secondary aerosols rather than primary emissions from combustion in this season. Levoglucosan, the tracer of biomass burning (Simoneit, 2002), also strongly correlated with fluorescent intensities (Figure S5d), suggesting that biomass burning was an important source of fluorescent BrC. The contribution of biomass burning to aerosol fluorophores is also suggested by previous studies (Qin et al., 2018; Xie et al., 2020; Dey et al., 2021)

Figure 3a presents typical EEM fluorescence spectra of water-soluble fluorophores in $PM_{2.5}$ samples in winter and summer in Tianjin, respectively. PARAFAC analysis on the basis of EEM spectra was conducted to provide more knowledge of the chemical composition and source of BrC fluorophores. Five independent fluorescent components in water-soluble BrC were identified by PARAFAC model with the total explained variance of 99.65 % within the whole sampling period (Figure 3b). Figure 3c presents the emission and excitation spectra of each component at peak emission and excitation wavelengths. The

fluorescent intensities of water-soluble BrC with relative abundances of each fluorescent component varied for different samples (Figure S6), suggesting that the chemical compositions of fluorescent organic matters were highly variable. The fluorophore C1 presents a primary fluorescent peak at Excitation/Emission (Ex/Em) of ~250 nm/ 395 nm and secondary peak at Ex/Em of ~315 nm/ 395 nm. C1 can be classified as a humic-like fluorophore because the bimodal distribution of fluorescence spectra is typically associated with humic-like substance (HULIS) (Coble, 2007; Murphy et al., 2011; Yu et al.,

2015). The second peak at the high excitation wavelength suggests there are plenty of condensed aromatic moieties, conjugated bonds and nonlinear ring systems (Matos et al., 2015). The fluorophore C2 exhibiting a peak at Ex/Em of ~250 nm/ 465 nm is also a humic-like fluorophore. The longer wavelength of C2 suggests that compared with C1, C2 is more aromatic with higher molecular weight, containing more conjugated and unsaturated chemical structures due to condensation reactions (Matos et al., 2015; Fan et al., 2020a; Dey et al., 2021). C3 showing a peak at Ex/Em of ~250 nm/ 385 nm is also a humic-like fluorophore

(Fan et al., 2020a; Li et al., 2020b). Based on previous research, it is inferred that that C1 and C3 both represent fluorophores





containing lowly-oxygenated organic species and C1 is more oxidized than C3; however, C2 is associated with highly-oxygenated structures (Elcoroaristizabal et al. 2014; Chen et al., 2016).

The other two fluorescent components (i.e., C4 and C5) present different spectral features from the humic-like fluorophores, and they are identified as protein-like fluorophores due to their short emission wavelengths (Coble, 1996, 2007). C4 showing
a fluorescence peak at Ex/Em of ~250 nm/ 340 nm is often associated with tryptophan-like fluorophore (Murphy et al., 2011, 2013). C5 with a peak at Ex/Em of ~275 nm/ 305 nm is generally regarded as a typical tyrosine-like fluorophore (Stedmon and Markager, 2005; Murphy et al., 2011). The significant correlations between BIX and C4 ($R = 0.583$, $p < 0.01$) or C5 ($R = 0.369$, $p < 0.01$) may support the possible contributions of bioaerosols to the protein-like compounds in the components (Figure S7a and S7b). It should be noted that due to the similar fluorescence spectra, the two protein-like fluorophores are also probably
related to some PAHs-like or phenol-like species from fossil fuel combustion and biomass burning, which is particularly in the case of urban aerosols (Elcoroaristizabal et al. 2014; Matos et al., 2015; Chen et al., 2020). For example, to a certain extent the spectrum of C4 is overlapped with that of naphthalene, an aromatic compound from fossil fuel combustion (Mladenov et al., 2011; Wu et al., 2019). Another evidence supporting the likely impacts of fossil-fuel combustion activities on the two protein-like components in BrC is the strong correlations of low molecular weight *n*-alkanes with both C4 ($R = 0.880$, $p < 0.01$)
and C5 ($R = 0.842$, $p < 0.01$) (Figure S7c and S7d), since low molecular weight *n*-alkanes were mainly derived from emissions of incomplete combustion of fossil fuels (Xie et al., 2009). Furthermore, C5 produces spectra similar to the fluorophore which may be related to non-nitrogen-containing species (Chen et al., 2016).

Correlation coefficients were further obtained between fluorescent intensities of each PARAFAC component and chemical compositions to identify the sources of different fluorophores. During the sampling periods, EC showed significantly strong
correlations with all the fluorescent components, except C5 in summer (Figure 4a1 and 4b), again providing support to the influence of primary emissions from combustion-related sources on BrC fluorophores. Similarly, the relationships between levoglucosan and BrC fluorophores suggest that biomass burning contributed to the formation of all the humic-like and protein-like fluorophores except C5 in summer (Figure 4c and 4d). Correlation between C5 and EC in summer was weak ($R = 0.198$, $p < 0.01$), and that between C5 and SOC was comparatively stronger ($R = 0.326$, $p < 0.01$), indicating that combustion processes
were not the dominant sources of C5 in summer and secondary formation even played a more important role. The considerable impacts of secondary formation on fluorescent BrC were indicated by the significant correlations between SOC and all the components, especially in winter (Figure 4e and 4f). It is noted that C2 presented the strongest correlation with SOC among the humic-like fluorophores, followed by C1 and C3. This finding supported the hypothesis that C2 is associated with highly-oxygenated structures, while C3 is less oxidized than the lowly-oxygenated component C1.
Figure 5 illustrate the average relative contributions of the fluorescent components for water-soluble BrC in different periods. On average, the humic-like fluorophores together contributed 73.4 % and 68.7 % to the fluorescence intensity in winter and summer, respectively, suggesting that humic-like fluorophores played a dominant role in fluorescence properties of water-soluble BrC in Tianjin. Generally, for winter samples, the lowly-oxygenated fluorophores C1 (27.5 %) and C3 (24.4 %) both made considerable contributions, followed by the highly-oxygenated fluorophore C2 (21.5 %). By contrast, in summer, relative



375 contribution made by the least-oxidized component C3 remarkably decreased to 8.6 %. Meanwhile, C1 presented much more abundant (35.8 %) and C2 also slightly increased (24.3 %). The larger relative contributions of more-oxygenated fluorophores in summer might partly attribute to the reason that the lowly-oxygenated fluorophores would be photodegraded through exposure to the strong summer solar radiation and then convert into more-oxygenated fluorophores through the oxygenation reaction pathways. This can further support the results that BrC were more aged in summer than in winter, which were

380 previously revealed by the fluorescent indices. In addition, this photoinduced mechanism can also explain the diurnal variations in relative abundances of humic-like fluorophores, which was characterized by smaller contribution of the lowly-oxygenated fluorophore C3 and larger contribution of the highly-oxygenated fluorophore C2 in daytime than nighttime in both seasons. Protein-like fluorophores were also vital components for fluorescent BrC since the two components together accounted for 26.6 % and 31.3 % of the total fluorescence intensities in winter and summer, respectively (Figure 5). One possible reason for

385 the larger contribution of protein-like fluorophores during summertime is the higher relative abundances of bioaerosols from fungal spores and plant debris due to the enhanced biological activities in summer, which can be supported by our previous research (Fan et al., 2020b). C4, the component significantly influenced by combustion sources, made a much smaller contribution in summer (12.4 %) compared with winter (20.7 %). However, C5 contributed 18.9 % in summer, more than three times larger than in winter (5.9 %). This is not surprising since C5 had fluorescent spectra similar to that of a photochemically-

390 formed fluorophore (Chen et al., 2020). Therefore, the larger contribution of C5 was likely due to the excessive oxidation and decomposition processes in summer, and this was particularly the case in daytime.

**3.5 Sources of BrC**

Relationships between BrC light absorption and carbonaceous species of aerosols were examined to investigate the sources of BrC in Tianjin (Figure 6a-d). Temporal pattern of $Abs_{365}$ was quite similar to that of WSOC (Figure S1a) and OC (Figure S1b).

395 The strongly positive correlations between $Abs_{365}$ and WSOC ($R = 0.837$–$0.947$, $p < 0.01$, Figure 6a) and OC ($R = 0.851$–$0.956$, $p < 0.01$, Figure 6b) indicates that the sources of BrC were similar to those of WSOC and OC. $Abs_{365}$ also well correlated with EC ($R = 0.695$–$0.789$, $p < 0.01$, Figure 6c), suggesting that the combustion-related processes were important sources of ambient BrC in both seasons, since EC is mainly from incomplete combustion of biomass and fossil fuels. We have found that biomass burning was one of the most abundant sources of OC in Tianjin, especially in winter (Fan et al., 2020b). $Abs_{365}$ was

400 significantly correlated with levoglucosan ($R = 0.498$–$0.665$, $p < 0.01$), the typical organic molecular tracer of biomass burning, indicating that a considerable fraction of BrC aerosols were associated with biomass burning activities in both winter and summer (Figure 6e). This is in accordance with previous findings that biomass burning is an important source of atmospheric BrC (Lack et al., 2012; Lin et al., 2016). Levoglucosan concentration in winter ($252 \pm 145$ µg m$^{-3}$) was more than 10 times larger than that in summer ($23.6 \pm 34.4$ µg m$^{-3}$), suggesting the much larger contribution of biomass burning to winter OC and

405 BrC (Table S1). The influences of biomass burning on BrC can also be supported by the strong linear correlations of $Abs_{365}$ with K$^+$ ($R = 0.784$–$0.789$, $p < 0.01$, Figure 6f), since K$^+$ is another tracer of biomass burning. However, the relatively weaker correlation between $Abs_{365}$ and levoglucosan in winter in comparison with that in summer indicates the relative contribution





of biomass burning might be smaller in winter. It is not surprising because combustion of fossil fuel such as coal and petroleum played a more important role in winter BrC formation due to the intense anthropogenic activities (e.g., heating) in cold season.

This can be supported by the much stronger linear correlations of $Abs_{365}$ with $SO_4^{2-}$ ($R = 0.812$, $p < 0.01$, Figure 6g) and $NO_3^-$ ($R = 0.769$, $p < 0.01$, Figure 6h) in winter, since the precursors of $NO_3^-$ (e.g., $NO_x$) and $SO_4^{2-}$ (e.g., $SO_2$) are mainly emitted from fossil-fuel combustion. Such results coincide with previous studies which found that fossil-fuel combustion made great contributions to winter aerosols in Tianjin (Huang et al., 2017; Gao et al., 2018).

Relationships between $Abs_{365}$ and molecular markers of organic aerosols from other specific emission sources were analyzed

to investigate the potential sources of atmospheric BrC. Besides primary emissions from combustions, bioaerosols, which contain various particle types such as bacteria, algae, pollen, fungal spores, plant debris and biopolymers, are also important sources of BrC aerosols (Andreae and Gelencsér, 2006; Pöhlker et al., 2013). Many sugar compounds are emitted persistently from biological sources and have been viewed as tracers of primary bioaerosols (Hu et al., 2020). For example, arabitol and mannitol are the major species in fungi and therefore used as the tracer for airborne fungal spores (Bauer et al., 2008); glucose

is predominantly derived from terrestrial vegetative fragments such as pollen, fruit, and debris (Pacini, 2000); trehalose, a metabolite of many microorganisms, is also frequently recognized as fungal carbohydrates (Simoneit, 2004); xylose, a monosaccharide, is the main component of hemicellulose in biomass and comes from different sources such as bacteria, vegetation, microbiota and biomass burning (Wan et al., 2019). Therefore, the significant correlations of $Abs_{365}$ with arabitol ($R = 0.419$–$0.494$, $p < 0.01$, Figure 6i), mannitol ($R = 0.407$–$0.422$, $p < 0.01$, Figure 6j), glucose ($R = 0.347$–$0.401$, $p < 0.01$,

Figure 6k), trehalose ($R = 0.306$–$0.489$, $p < 0.01$, Figure 6l) and xylose ($0.476$–$0.773$, $p < 0.01$, Figure 6m) suggest bioaerosols also contributed to BrC in Tianjin. The potential impacts of bioaerosols on BrC can be supported by the PARAFAC-derived fluorescent results as discussed in Sect. 3.4.

Moreover, $Abs_{365}$ and the estimated SOC showed similar variations with strong correlations in both summer ($R = 0.693$, $p < 0.01$) and winter ($R = 0.881$, $p < 0.01$) (Figure 6d and S1d), suggesting the significant impacts of secondary formation processes

on BrC, even in the winter season. This is reasonable because SOA are also major contributors to BrC through photochemical reactions and/or aqueous/heterogeneous chemical processes (Laskin et al., 2014; Lin et al., 2015; Braman et al., 2020; Kasthuriarachchi et al., 2020). The correlation between $Abs_{365}$ and SOC in winter was even stronger than that between $Abs_{365}$ and EC, indicating the greater impacts of secondary formation on winter BrC compared with the primary emissions from combustion-related activities. The large contribution of secondary formation to wintertime BrC is also indicated by the strong

correlations of BrC with $SO_4^{2-}$ and $NO_3^-$ (Figure 6g–h). To explore the relationships between BrC and SOA formed through different reaction pathways, correlations of $Abs_{365}$ with specific molecular markers of biogenic SOA tracers (e.g., $C_5$-alkene triols, 2-methyltetrols (MTLs), 2-methylglyceric acid (2-MGA), pinic acid, and 3-hydroxyglutaric acid (3-HGA)) and anthropogenic SOA tracers (e.g., 2, 3-dihydroxy-4-oxopentanoic acid (DHOPA), and phthalic acids) were also examined.

The ubiquitous biogenic volatile organic compounds (BVOCs) are believed to contribute to the formation of atmospheric BrC

by complex atmospheric processes (Bones et al., 2010; Updyke et al., 2012; Nguyen et al., 2013; Laskin et al., 2015). $C_5$-alkene triols and MTLs are the major SOA tracers due to isoprene photooxidation under low-$NO_x$ conditions, and 2-MGA is





a further oxidation product of isoprene under high-NO$_x$ conditions; Pinic acids is the first-generation oxidation product of monoterpene, while 3-HGA is the higher generation product (Kang et al., 2018; Fan et al., 2020b). Figure 6n–r demonstrates the significant correlations between Abs$_{365}$ and C$_5$-alkene triols ($R = 0.395$–$0.469$, $p < 0.01$), MTLs ($R = 0.476$, $p < 0.01$ in

summer), 2-MGA ($R = 0.342$–$0.514$, $p < 0.01$), pinic acid ($R = 0.231$–$0.421$, $p < 0.01$), and 3-HGA ($R = 0.388$–$0.415$, $p < 0.01$). These weak to moderate correlation coefficients suggest that the biogenic SOA formation processes are potential sources of BrC in Tianjin although their contributions might be not large. Since the contributions of biogenic SOA to OC significantly elevated in summer due to the larger emissions of their precursors and strong photooxidation (Fan et al., 2020b), biogenic SOA might be an important source of BrC in summer. DHOPA and phthalic acids are tracers for the anthropogenic SOA from

toluene and naphthalene, respectively (Kleindienst et al., 2012; Fu et al., 2014). Figure 6s–t shows that Abs$_{365}$ exhibited significantly strong relevancies with DHOPA ($R = 0.472$–$0.638$, $p < 0.01$) and phthalic acids ($R = 0.400$–$0.742$, $p < 0.01$), again confirming that anthropogenic SOA formation is also an important source for BrC in Tianjin. In addition, the much larger correlation coefficients in winter indicate the stronger influence of anthropogenic activities on secondary formation of winter BrC, in accordance with the results suggested by the stronger correlations of winter BrC with SO$_4^{2-}$ and NO$_3^-$.

The potential sources of BrC in Tianjin with their relative contributions were further analyzed with the PMF model constrained by organic molecular markers. Figure 7 presents the profiles of the five factors, together with the temporal variations in contributions from individual factors. The first factor (F1) is mainly related to anthropogenic SOA formation processes since it is characterized by the highest level of DHOPA. The second factor (F2), which is featured by the high abundances of glucose, trehalose, mannitol and arabitol, is primarily derived from bioaerosol emissions. The third factor (F3), with the largest

contributions of levoglucosan and its two isomers (i.e., galactosan and mannosan), is identified as the source of biomass burning, since galactosan and mannosan can also act as biomass burning tracers (Simoneit, 2002). The fourth factor (F4) shows highest loadings of C$_5$-alkene triols, MTLs, 2-MGA, and pinic acid, and therefore, it can be identified as the source from biogenic SOA formation. The fifth factor (F5) had the largest abundances of SO$_4^{2-}$, NO$_3^-$, NH$_4^+$, K$^+$, and Cl$^-$. Therefore, F5 is likely to associate with the fossil fuel combustion and aging processes.

Mean relative contributions of various sources to BrC in Tianjin in different periods are plotted in Figure 8. Obvious diurnal changes were found in source contributions of BrC. Bioaerosol emission made a larger contribution to BrC at daytime than at nighttime, especially in summer, suggesting the influences of daytime biological activities on BrC formation. Biomass burning contributed more to BrC at night, which called for more attentions to nighttime BrC formation due to biomass-burning activities. In addition to the diurnal differences, remarkable seasonal changes in sources of BrC also existed. In winter, biomass burning

as well as fossil fuel combustion and aging processes were the predominant sources of BrC, with the contributions of 30.7% and 30.0%, respectively. The predominant contributions of biomass burning and fossil fuel combustion to winter BrC in Tianjin were coincided with the source apportionment results of BrC in Xi'an in northwest China (Wu et al., 2020a; Yuan et al, 2020). Anthropogenic SOA formation and bioaerosol emission also made large contributions to BrC in winter, accounting for 21.1% and 18.2%, respectively. Biogenic SOA formation made little contribution to winter BrC, which might be due to the weak

emissions of BVOCs under low temperature.



However, in summer, biogenic SOA formation accounting for 23.7% became a prominent source of BrC. Primary bioaerosol emission (38.1%) was found to be the most important source of summertime BrC, which was led by the strong biological activities. Anthropogenic SOA formation also played an important role although its relative contribution in summer (19.8%) was slightly smaller than that in winter, suggesting the considerable significance of anthropogenic secondary BrC in both seasons. Compared to the significant influence in winter, fossil fuel combustion and aging processes in summer played a much minor role in BrC formation because its relative contribution dramatically decreased to 15.3%. Besides, biomass burning made a trivial contribution of 3.2% to summer BrC, which contrasted sharply to the dominant contribution in winter. To sum up, the combustion-related primary emissions played a much more important role in BrC formation in winter, while secondary formation from photochemical and aqueous chemical processes contributed more to BrC in summer than in winter. Such source apportionment results again indicating that BrC in Tianjin were more affected by fresh emission and less aged in winter while more aged in summer, in accordance with the fluorescent results. The larger contributions of secondary BrC may also partly explain the lower MAE in summer (Table 1), since atmospheric aging processes would weaken light absorption (Zhong and Jang, 2014; Liu et al., 2016).

**4 Conclusions**

This study presents the temporal variations in light absorption and fluorescent properties of water-soluble BrC in PM$_{2.5}$ over Tianjin in North China in winter and summer during 2016–2017. Sources of water-soluble BrC were comprehensively analyzed by investigating the relationships of BrC and chemical compositions of aerosols. Results show that light absorption properties of BrC experienced obvious seasonal variations. Abs$_{365}$, AAE, MAE$_{365}$, and $k_{365}$ of BrC were 6.8, 1.1, 1.8, and 1.8 times larger in winter than in summer, respectively, suggesting the much more abundant and stronger light absorbing of water-soluble BrC in winter. However, there are no significant differences in BrC light absorption between daytime and nighttime. Water-soluble BrC contributed significantly to absorption radiative forcing, especially in the UV range, indicating their considerable influences on climate warming and ozone formation. Fluorescent indices present that BrC were associated with a slightly higher aromaticity degree and polycondensation in their chemical structures in summer and daytime, which was likely resulted from photobleaching processes. Three humic-like components (C1, C2, and C3) and two protein-like components (C4 and C5) were determined as the major fluorescent organics by PARAFAC analysis. The humic-like components were predominant in both seasons and their relative contributions were larger in winter than in summer. The lowly-oxygenated humic-like components contributed more in winter and nighttime, while relative contributions of more-oxygenated humic-like components obviously increased in summer and daytime, indicating the less-oxygenated fluorophores might be oxidized into more-oxygenated fluorophores due to photodegradation. Combustion processes and secondary formation remarkably contributed to the humic-like and protein-like fluorescent organic aerosols. Correlation analysis between BrC and chemical compositions of aerosols as well as source apportionments by PMF analysis with the organic molecular tracers suggested that sources contributions of BrC presented obvious seasonal and diurnal variations and fossil fuel



combustion and aging processes, bioaerosol emission, and anthropogenic SOA formation were important sources of BrC in both winter and summer. Impacts of biomass burning in winter, and primary biological aerosol emission and biogenic
secondary formation in summer on BrC were highlighted by the PMF results. Overall, source contributions and fluorescent properties together indicated that BrC were prominently affected by freshly emitted aerosols and less aged in winter, while they were more aged in summer. This study broadens our knowledge of optical properties, sources, and evolution formation of BrC in the heavily polluted urban region in China, which will help to estimate climatic effect of atmospheric aerosols and control carbonaceous aerosol pollution.


*Data availability.* The data is available upon request from the corresponding author.

*Author contribution.* PF designed the experiments. JD, XW, SZ, and ZZ carried out the experiments and performed the data analysis. JD prepared the manuscript with contributions from all co-authors.

*Competing interests.* The authors declare that they have no conflict of interests.

*Acknowledgments.* This study was supported by the National Key Research and Development Program of China (2019YFA0606801), Strategic Priority Research Program of the Chinese Academy of Sciences (No. XDA23020301), National Natural Science Foundation of China (21607148, 41625014), Natural Science Foundation of Tianjin City (20JCQNJC01590), and Peiyang Young Scholar Program of Tianjin University (2020XRG-0068).

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



**Figure 1. Temporal variations in light absorption and fluorescence properties of BrC in Tianjin: (a) AAE, (b) Abs$_{365}$, (c) MAE$_{365}$, (d) SFE, (e) $f$, (f) HIX, (g) BIX, and (h) FI.**






**Figure 2. Scatter plots of the HIX values as a function of (a) FI and (b) BIX for WSOC in Tianjin aerosols, and comparison plots of HIX with (c) FI and (d) BIX in aerosols in this study and literature.**




**Figure 3. (a) Typical excitation-emission matrix (EEM) fluorescence spectra of water-soluble organic carbon in the aerosol samples collected in winter and summer, respectively. (b) Three-dimensional excitation-emission matrix of five fluorescent components (C1–C5) in BrC obtained by PARAFAC model analysis. (c) Emission and excitation spectra of each fluorescent component at peak emission and excitation wavelengths.**




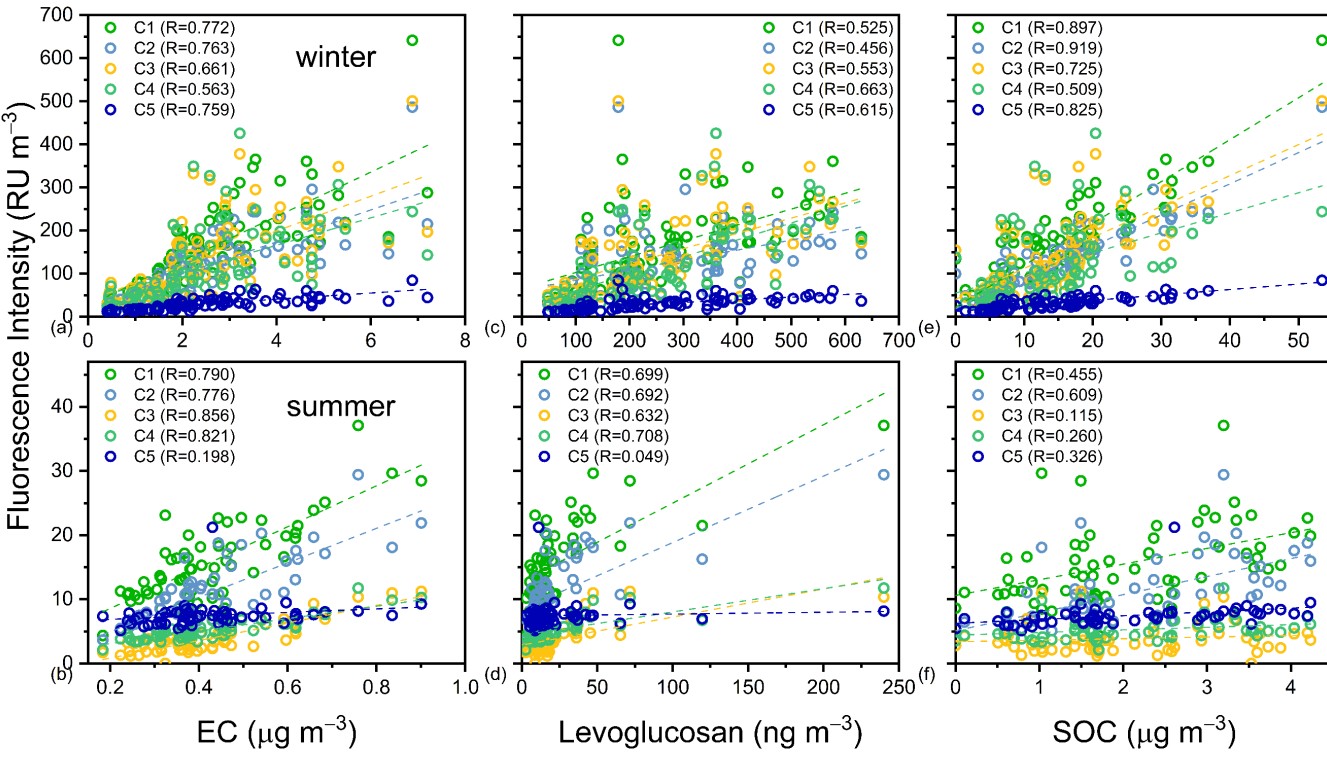

**Figure 4. Correlations of fluorescent intensities of five fluorescent components in BrC with EC (a–b), levoglucosan (c–d), and SOC (e–f) in winter and summer.**





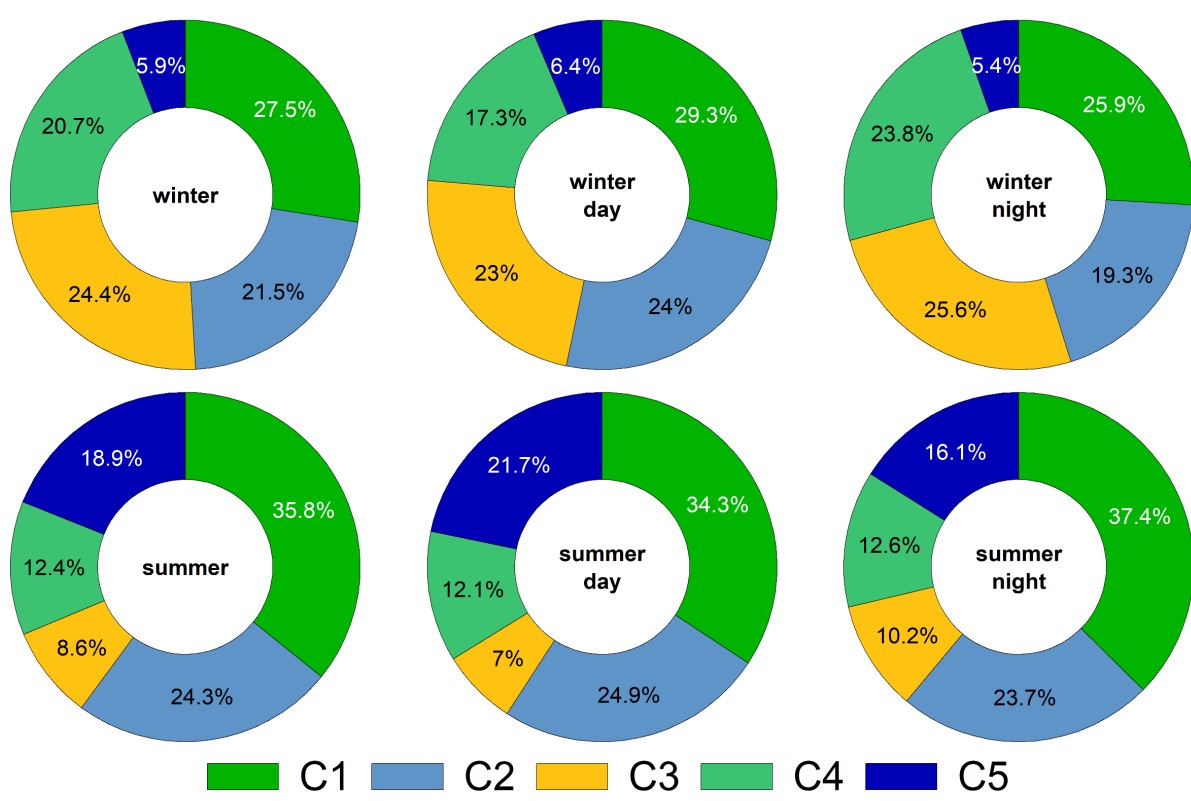

**Figure 5. Average relative abundances of the PARAFAC-derived fluorescent components for water-soluble BrC of Tianjin aerosols in different periods. C1, C2 and C3 are humic-like components, and C4 and C5 are protein-like components.**



**Figure 6. Relationships between Abs$_{365}$ and chemical species of aerosols in Tianjin: (a) WSOC, (b) OC, (c) EC, (d) SOC, (e) levoglucosan, (f) K$^+$, (g) SO$_4^{2-}$, (h) NO$_3^-$, (i) arabitol, (j) mannitol, (k) glucose, (l) trehalose, (m) xylose, (n) C$_5$-alkene triols, (o) 2-methyltetrols (MTLs), (p) 2-methylglyceric acid (2-MGA), (q) pinic acid, (r) 3-hydroxyglutaric acid (3-HGA), (s) 2,3-dihydroxy-4-oxopentanoic acid (DHOPA), and (t) phthalic acids.**



Figure 7. (a) The left figure shows the individual source profiles of the factors resolved by PMF analysis, and the right one (b) shows the temporal variations in individual factor contributions to BrC.





**Figure 8. Relative contributions of individual sources to BrC obtained by PMF analysis with molecular marker.**




**Table 1. Light-absorbing and fluorescence properties of BrC in PM$_{2.5}$ in urban Tianjin, China.**

| | Summer | | | Winter | | |
|---|---|---|---|---|---|---|
| | Day (N=30) | Night (N=30) | Average (N=60) | Day (N=41) | Night (N=43) | Average (N=84) |
| *Light absorption property* | | | | | | |
| Abs$_{365}$ (Mm$^{-1}$) | 2.0±0.8 | 2.1±1.1 | 2.1±1.0 | 14.4±10.3 | 13.9±6.3 | 14.1±8.5 |
| MAE$_{365}$ (m$^2$ g$^{-1}$) | 0.80±0.21 | 0.88±0.24 | 0.84±0.22 | 1.50±0.33 | 1.58±0.33 | 1.54±0.33 |
| AAE | 4.8±0.6 | 4.9±0.6 | 4.9±0.6 | 5.4±0.4 | 5.4±0.3 | 5.4±0.4 |
| E$_{250}$/E$_{365}$ | 5.7±0.7 | 5.6±0.7 | 5.7±0.7 | 5.7±0.6 | 5.6±0.6 | 5.6±0.6 |
| $k_{365}$ | 0.040±0.010 | 0.043±0.012 | 0.042±0.011 | 0.074±0.016 | 0.078±0.016 | 0.076±0.016 |
| $f_{300\text{-}400}$ (%) | 47.5±13.9 | 41.7±13.6 | 44.6±13.9 | 56.2±16.8 | 52.6±17.0 | 54.3±16.9 |
| $f_{280\text{-}4000}$ (%) | 13.4±4.5 | 11.7±4.4 | 12.5±4.5 | 14.0±4.0 | 13.1±4.3 | 13.5±4.1 |
| SFE$_{300\text{-}700}$ (W g$^{-1}$) | 4.8±1.8 | 4.4±1.6 | 4.6±1.7 | 6.3±2.3 | 6.0±1.6 | 6.2±2.0 |
| SFE$_{300\text{-}400}$ (W g$^{-1}$) | 1.3±0.4 | 1.4±0.4 | 1.4±0.4 | 2.3±0.5 | 2.4±0.5 | 2.4±0.5 |
| *Fluorescence property* | | | | | | |
| FI | 1.58±0.09 | 1.65±0.08 | 1.61±0.10 | 1.67±0.04 | 1.74±0.06 | 1.71±0.06 |
| BIX | 1.13±0.12 | 1.25±0.10 | 1.19±0.13 | 1.25±0.10 | 1.39±0.13 | 1.32±0.14 |
| HIX | 2.59±0.54 | 2.86±0.45 | 2.73±0.51 | 2.55±0.44 | 1.91±0.43 | 2.22±0.54 |
