# Peer review of "Measurement Report: Optical properties and sources of water-soluble brown carbon in Tianjin, North China: insights from organic molecular compositions"

_Atmospheric Chemistry and Physics, 2021_

## Author Comment (AC1)

**Reply to Reviewer #1**

This manuscript investigated the optical properties, sources and radiative impacts of water-soluble brown carbon in Tianjin, a representative megacity in the North China Plain. Daytime and nighttime samples were collected during winter and summer, and analyzed for aerosol compositions, light absorption spectra and fluorescence properties. Based on the measurement results, connections between the chemical compositions and optical characteristics of aerosols were explored; in addition, radiative impacts of brown carbon were estimated using different approaches. My overall assessment is that this manuscript could be considered for publication as a Measurement Report after a minor revision. My detailed comments are given below.

Reply: We appreciate for the positive comments and helpful suggestions. We have carefully revised our manuscript accordingly. We list our detailed reply in the following sections in blue text.

Line 16. Suggest change "aerosols" to "particles".

Reply: Changed accordingly.

Lines 19-20. Check this sentence.

Reply: This sentence was modified as follows:

"In winter, BrC showed much stronger light absorbing ability, since mass absorption efficiency at 365 nm ($MAE_{365}$) in winter ($1.54 \pm 0.33$ m$^2$ g$^{-1}$) was 1.8 times larger than $MAE_{365}$ in summer ($0.84 \pm 0.22$ m$^2$ g$^{-1}$)."

Line 21. I guess something was missing after "$44.6 \pm 13.9$ %", e.g., were the two values for different seasons?

Reply: We have modified this sentence as follows:

"Direct radiative effects by BrC absorption relative to black carbon in the UV range were $54.3 \pm 16.9$ % and $44.6 \pm 13.9$ % in winter and summer, respectively."

Line 45. It should be "On one hand".

Reply: Sorry for the typo. It has been corrected.

Line 48. Secondary BrC could also be formed through gas-phase (i.e., photochemical) reactions.

Reply: We agree to the comment and have modified this sentences to "On the other hand, BrC can be formed through gas-phase, aqueous-phase or heterogeneous reactions from both biogenic and anthropogenic precursors."

Section 2.1 The number of samples should be clarified. This information is important for the reliability of the PMF results.

Reply: Totally, 84 winter samples and 60 summer samples were collected during this campaign. Accordingly, this information has been added in the revised manuscript (Line 97).

Lines 107-108. Details on the EC-tracer method should be provided, e.g., determination of the OC/EC ratio representative of primary emissions.

Reply: The brief introduction of the method has been implemented accordingly as follows:

"Secondary organic carbon (SOC) was estimated with the EC tracer method, which assumed that in all samples the OC/EC ratio for the primary sources affecting the site remains constant (Castro et al., 1999):

$$SOC = OC - \left[ EC \times (OC/EC)_{min} \right]$$

where $(OC/EC)_{min}$ is the minimum value of OC/EC ratios in each season."

Line 133 and elsewhere in the manuscript. Check the unit of MAE (m$^2$/g or m$^2$/gC).

Reply: The unit of MAE should be m$^2$ gC$^{-1}$. We revised the units throughout the manuscript.

Equation (4). Check whether the WSOC mass has been converted to that of water-soluble organic matter.

Reply: In Eq.4, we used the mass concentration of WSOC rather than water-soluble organic matter according to previous studies (Liu et al., 2013; Shamjad et al., 2018).

References:

Liu, J., et al.: Size-resolved measurements of brown carbon in water and methanol extracts and estimates of their contribution to ambient fine-particle light absorption, Atmos. Chem. Phys., 13, 12389–12404, 2013.

Shamjad, P. M., et al.: Absorbing Refractive Index and Direct Radiative Forcing of Atmospheric Brown Carbon over Gangetic Plain, ACS Earth Space Chem., 2, 31–37, 2018.

Lines 197-199. Was the difference in AAE statistically significant?

Reply: Yes. The difference in AAE was statistically significant ($p < 0.01$) based on pair-sample $t$-test.

Lines 240-243. Were these values calculated by the same methodology?

Reply: Yes. The results were adopted from the limited available references using the same methodology.

Line 315 and elsewhere in the manuscript. Maybe it is better to use "r" (in italic) instead of "R".

Reply: We have used $r$ to replace R in the manuscript according to your suggestion.

Lines 403-404. Check the units of levoglucosan.

Reply: Sorry for the typo. We have modified the units to "ng m$^{-3}$".

Section 3.5. I would like to see the relationship between SOC estimated by the EC-tracer method and that derived from PMF analysis.

Reply: We calculate the contributions of different sources to BrC light absorption (Abs) rather than OC from PMF analysis. Therefore, according to the suggestion, we investigate the

relationship between Abs from SOC ($Abs_{SOC}$) derived from the PMF results and SOC estimated by the EC tracer method. As shown in the Figure, there is a strongly positive correlation between $Abs_{SOC}$ and SOC with correlation coefficient $r$ of 0.868.

[Figure]

Fig. R1. Correlations between light absorption from SOC ($Abs_{SOC}$) derived from the PMF results and SOC concentration estimated by the EC tracer method.

---

## Author Comment (AC2)

**Reply to Reviewer #2**

This paper reports on measurements of PM2.5 water-soluble BrC and fluorescent properties of these aerosol particles extracted from filters. Measurements of other species, including PM2.5 organic carbon, elemental carbon, water-soluble organic carbon and a number of specific organic species are used to determine the sources of the BrC. The measurements are made in a large city in China, Tianjin, and contrasts are made between winter and summer. There are a number of limitations and confusing aspects that should be clarified. First, it should be made very clear in the abstract and throughout the paper that the BrC discussed is light absorption of species in the water extract of filters collecting PM2.5 particles and that no consideration is given to conversion from solvent absorption to aerosol particle absorption (particle size effects on light absorption) and that insoluble BrC species are not included. Thus, the measurements and the analysis of BrC are not comprehensive and light absorption coefficients are not those of actual aerosol particles, which affects the calculations of the radiative effects and associated comparisons with BC. The utility of the fluorescence measurements in relation to BrC is not clear. Even on its own, why is fluorescence of interest; I assume it is to identify sources and processing of the aerosol? The authors seem to assert that there are specific organic species that absorb light and are fluorescent, which they refer to as BrC fluorophores, or fluorescent BrC. This may be so, but the data does not prove this since it is only based on a correlation analysis, whereas a single particle analysis is needed to show individual particles, or species with individual particles, have this property. I believe the authors are saying that high correlation between measurements of BrC and fluorescents provides insights on the sources and processing of species that contribute to BrC, but their discussion and terminology is confusing. These Issues should be addressed before publication.

Reply: Thank you so much for the helpful comments. We carefully revised our manuscript to according to your suggestions. We have clarified in the abstract and main text that the BrC discussed in this study were the water-soluble BrC, which were collected on the filters and then extracted by water. In addition, we revised the discussion of the fluorescence properties of water-soluble BrC. The following list our point-to-point responses to the specific comments with blue color.

Specific Comments:

In the Abstract state how this BrC was determined. That is, the BrC is only the water-soluble fraction and is measured in a solution extract, which is not aerosol BrC.

Reply: We have added the statements according to the suggestion as follows:

"Light absorption and fluorescence properties of water extracts of $PM_{2.5}$ were investigated to obtain seasonal and diurnal patterns of atmospheric water-soluble BrC".

In the Introduction the assertion that BrC has a large effect on the global radiation balance is based on some highly speculative modeling studies. One might consider studies based more on actual measurements.

Reply: According to the suggestions, some literatures (Kirillova et al., 2014; Lu et al., 2015; Zhu et al., 2021) based on field measurements have been added to support the effect of BrC on radiation balance.

References:

Kirillova, E. N., et al.: Water-soluble organic carbon aerosols during a full New Delhi winter: Isotope-based source apportionment and optical properties, J. Geophys. Res. -Atmos., 119, 3476–3485, 2014.

Lu, Z., et al.: Light absorption properties and radiative effects of primary organic aerosol emissions, Environ. Sci. Technol., 49, 4868–4877, 2015.

Zhu, C. S., et al.: Black Carbon and Secondary Brown Carbon, the Dominant Light Absorption and Direct Radiative Forcing Contributors of the Atmospheric Aerosols Over the Tibetan Plateau, Geophys. Res. Lett., 48, 2021.

Regarding the radiative forcing calculations, what is the limitation of assuming what is measured only at the surface can be used to predict top of atmosphere radiative properties? For example, can this prediction actually be compared to a model or measurements that considered BrC throughout the whole atmospheric column? Is there no light absorbing aerosol above the boundary layer? Also, only water-soluble BrC in a solution was measured, which is not comprehensive and not necessarily the same as BrC in the aerosol particles. Discuss how these various factors affect the prediction on the radiative importance of BrC (seems like they will result in a significant underestimation of BrC direct radiative effects). Also, in section 3.2, make sure the data being compared from this work to that of others involves the same approximations as used here (e.g., only water-soluble species considered, BrC of solution not aerosol, only BrC in the boundary layer). Overall, the direct radiative effects presented will likely not accurately estimate the actual impact of BrC species on TOA forcing.

Reply: We agree with you that the direct radiative effects presented in this study did not accurately represent actual impact of aerosol BrC on TOA forcing. In this work, the radiative forcing by BrC relative to BC were estimated with a widely-used simplistic model, which has been adopted in many previous studies (Bocsh et al., 2014; Kirillova et al., 2014, 2016; Bikkina and Sarin, 2019; Yue et al., 2019). The results being compared from this work to those of others in section 3.2 do adopt the same simplistic model and involve the same approximations (i.e., water-soluble BrC in the boundary layer).

As discussed in Bosch et al (2014), the simplistic estimate is based on the assumptions which may lead to uncertainty in the radiative forcing: (1) The light-absorptive properties of the solvent extracts are representative of the ambient aerosol phase and there are no effects of scattering or size distributions of aerosols; (2) The measurements at ground level represent a near surface well-mixed vertical aerosol layer with a height defined by atmospheric boundary layer. Our work aimed to characterize the optical properties of water-soluble BrC, and therefore we actually estimated the radiative effect of water-soluble BrC rather than all BrC species. In other words, our results only represent the water-soluble portion of BrC, while the estimated direct radiative effect should be less than the actual effect of aerosol BrC since the radiative effects of water-insoluble species were not considered.

In addition, we estimated BrC radiative effect in the boundary layer rather than the whole atmospheric column, also leading to some uncertainty in the radiative forcing, since there were BrC aerosols above the boundary layer. Due to the lack of vertical measurements of aerosols (or BrC) in the region of interest, the absorbing effect of BrC in the upper air was not included in

the estimation of radiative forcing with the simplistic model. This calculation is to broadly evaluate the solar spectrum integrated relative absorbance of Water-soluble BrC and BC for a ground-level situation. There is no implication of a column-integrated relative radiative effect of Water-soluble BrC versus BC, which are likely to have a nonhomogeneous vertical distribution. However, according to a previous study on aerosol vertical distribution from long-term satellite and ground-based remote sensing (Tian et al., 2017), about 80% of the column aerosols over the North China Plain are distributed within 1.5 km above the ground while light extinction by aerosols over 4 km above ground is much smaller (Fig. 2 and Fig. 10 of Tian et al., 2017).

In summary, we have added a discussion in the revised manuscript (Lines 265–271):

"It is stressed that the results in the present work only represent the direct radiative effects of water-soluble fractions of BrC, while the estimated effects should be less than the actual effects of aerosol BrC since the water-insoluble species in BrC were not considered in the calculation. In addition, due to the lack of vertical measurements of aerosols in the upper air, these estimated radiative effects reflect the solar spectrum integrated relative absorbance of water-soluble BrC and BC for a ground-level situation, and do not accurately represent actual impact of aerosol BrC on top-of-atmosphere radiative forcing. This assumption will lead to some uncertainty, although most of the column aerosols over the NCP are distributed within 1.5 km above the ground while light extinction by aerosols over 4 km above ground is much smaller (Tian et al., 2017)."

References:

Bikkina, S., and Sarin, M.: Brown carbon in the continental outflow to the North Indian Ocean, Environ. Sci.: Processes Impacts, 21, 970, 2019.

Bocsh, C. et al.: Source-diagnostic dual-isotope composition and optical properties of water-soluble organic carbon and elemental carbon in the South Asian outflow intercepted over the Indian Ocean, J. Geophys. Res., 119, 11743–11759, 2014.

Kirillova, E. N., et al.: Water-soluble organic carbon aerosols during a full New Delhi winter: Isotope-based source apportionment and optical properties, J. Geophys. Res. Atmos., 119, 3476–3485, 2014.

Kirillova, E. N., et al.: Light absorption properties of brown carbon in the high Himalayas, J. Geophys. Res. Atmos., 121, 9621–9639, 2016.

Tian, P., et al.: Aerosol vertical distribution and optical properties over China from long-term satellite and ground-based remote sensing, Atmos. Chem. Phys., 17, 2509–2523, 2017.

Yue, S., et al.: Sources and radiative absorption of water-soluble brown carbon in the high Arctic atmosphere, Geophys. Res. Lett., 46, 14881–14891, 2019.

Line 179, change liner to linear.

Reply: Corrected.

Line 313 states: "To explore the possible sources and controlling factors of fluorescent BrC", Are the authors really looking at specific organic species that both absorb light and are fluorescent? For example, below this line it states that there is a strong correlation between fluorescent intensities and EC. Does that mean EC contains organic fluorescent compounds? The line "a majority of light-absorbing BrC aerosols were fluorescent (Figure S5a)" is not proven by a correlation and terms like fluorescent BrC and BrC chromophores are very confusing, and in my view not accurate. Also, why does one care about these species? It seems the main reason for the interest in the fluorescent properties is that it provides information on the BrC,

and other species, sources. But that is not what line 313 states, which gives the impression the authors are looking specifically at organic species that absorb and are fluorescent, but which the data cannot prove exist. To summarize, I do not see proof for the existence of so-called BrC fluorophores. I do agree that one can find both of these types of species in a group of organic compounds, like HULIS, but that does not mean they are the same organic species. The authors should clarify this throughout and the whole section, (3.4 Fluorescent Components of BrC), which should be reviewed carefully and modified.

Reply: We agree with you that the BrC compounds (i.e., light-absorbing organic species) and the fluorescent compounds in aerosols are not exactly the same. Generally, the fluorescent compounds belong to BrC because they can absorb light, but some BrC compounds are not fluorescent. We need to clarify that in this work we did not focus on specific species of brown carbon that is both light-absorbing and fluorescent. By investigating fluorescent properties of BrC, we aimed to explore the possible sources and evolution processes of BrC, because the fluorescence properties may provide some information on BrC, such as compositions (e.g., humic-like compounds, and protein-like compounds) or oxidation degree.

By conducting correlation analysis of fluorescence intensity with other parameters, we investigated the potential sources of the fluorescent compounds in BrC (i.e., fluorophore). Therefore, the strong correlation between fluorescent intensities and EC does not mean EC contain fluorescent compounds. It only suggests that EC and fluorophores may have similar sources, such as combustion. According to your suggestions, in section 3.4 of the revised manuscript, we have changed the statements to avoid confusing and misleading. The title of this section was revised to "Compositions of BrC Identified by Fluorescence Analysis". In addition, the terms such as fluorescent BrC and BrC chromophores were deleted accordingly and BrC fluorophore was adopted to refer to the fluorescent compounds in BrC throughout the revision.

Ling 415 states: Besides primary emissions from combustions, bioaerosols, which contain various particle types such as bacteria, algae, pollen, fungal spores, plant debris and biopolymers, are also important sources of BrC aerosols (Andreae and Gelencser, 2006; Pohlker et al., 2013). Is this really that important for $PM_{2.5}$, many of these species are most likely in the coarse mode? Are these source of water-soluble BrC (I doubt, pollen, spores and plant debris are soluble), which is important since the authors are measuring only water-soluble BrC. Same questions apply to the discussion after the line copied above. The authors should clarify if they are mostly talking about BrC of coarse or fine particles here and are they water-soluble.

Reply:

(1) on the size of bioaerosols

In fact, it has been revealed by many studies that primary biological aerosol particles (PBAPs) are distributed in a wide size range from ~ 1 μm to > 10 μm (Fröhlich-Nowoisky et al., 2016). Gabey et al (2010) analyzed the number concentration of fluorescent primary biological aerosol particles (FBAPs) below and above a tropical rainforest canopy, suggesting that FBAPs mainly distributed in the size range of 1–10 μm and peaked around 2.5 μm. Savage et al (2017) investigated the size distributions of bioaerosols such as pollen, fungi and bacteria, and found that bacteria and pollen obviously distributed in the fine mode with diameter < 2 μm. Yue et al (2017) found that primary bioaerosols in urban Beijing showed two size modes which were around 2–4 μm and 1 μm, respectively. Bioaerosols in Tianjin were also significantly contributed

by particles around 1 μm, which were supposed to be most likely to be bacteria (Cheng et al., 2020). In addition, according to our previous study on the molecular compositions for the PM$_{2.5}$ samples analyzed in this work, the fine particles in Tianjin were apparently influenced by biological sources, especially in summer (Fan et al., 2020). Therefore, although biological aerosols are more likely in the coarse mode, they are also frequently distributed in fine particles.

(2) on the contribution of water-soluble BrC from bioaerosol

We note that although many biological aerosols are water-insoluble, some bioaerosol-related components (or biological markers) are water-soluble and light-absorbing (i.e., BrC), such as some proteins and amino acids. Amino acids can be emitted from biogenic sources or degraded from proteinaceous substances. Atmospheric amino acids account for a large fraction of water-soluble organic nitrogen compounds in aerosols (Hu et al., 2020). Direct photoinduced damage to proteins by UV light is limited to a certain number of amino acid residues such as tyrosine and tryptophan (Hu et al., 2021). In addition, degradation of bioaerosols will change their physicochemical properties. For example, airborne bioaerosols can be degraded by photooxidation of proteinaceous matters and contribute to HULIS during the long-range transport (Yue et al, 2019). And as we know, HULUS contains water-soluble BrC components. Therefore, we believe bioaerosols (or bioaerosol-related components) will contribute to atmospheric water-soluble BrC.

Therefore, in the revised manuscript, the text has been rewritten as follows (Lines 423–430):

"Relationships between Abs$_{365}$ and molecular markers of organic aerosols from other specific emission sources were analyzed to investigate the potential sources of atmospheric BrC. Besides primary emissions from combustions, bioaerosols, which contain various particle types such as bacteria, algae, pollen, fungal spores, plant debris and biopolymers, are also important sources of BrC aerosols (Andreae and Gelencser, 2006; Pohlker et al., 2013). With number and mass concentration in the size range with diameters > ~ 1 μm, bioaerosols make significant contributions to atmospheric aerosols (Fröhlich-Nowoisky et al., 2016). Although many bioaerosols are water-insoluble, some bioaerosol-related components (or biological markers) are water-soluble and light-absorbing (i.e., water-soluble BrC), such as some proteins and amino acids. For example, being emitted from biogenic sources or degraded from proteinaceous substances, atmospheric amino acids account for a large fraction of water-soluble organic nitrogen compounds in aerosols (Hu et al., 2020)."

References:

Cheng, B., et al.: Summertime fluorescent bioaerosol particles in the coastal megacity Tianjin, North China, Sci. Total Environ., 723, 137966.

Fan, Y., et al.: Large contributions of biogenic and anthropogenic sources to fine organic aerosols in Tianjin, North China, Atmos. Chem. Phys., 20, 117–137, 2020.

Frohlich-Nowoisky, J., et al.: Bioaerosol in the Earth system: Climate, health, and ecosystem interactions, Atmos. Res., 182, 346–376, 2016.

Gabey, A. M., et al.: Measurements and comparison of primary biological aerosol above and below a tropical forest canopy using a dual channel fluorescence spectrometer, Atmos. Chem. Phys., 10, 4453–4466, 2010.

Hu, W., et al.: Biological Aerosol Particles in Polluted Regions, Curr. Pollut. Rep., 6, 2, 65–89, 2020.

Hu, W., et al.: Photochemical Degradation of Organic Matter in the Atmosphere, Adv. Sustainable Syst., 5, 2100027, 2021.

Savage, N. J., et al.: Systematic characterization and fluorescence threshold strategies for the wideband integrated bioaerosol sensor (WIBS) using size-resolved biological and interfering particles, Atmos. Meas. Tech., 10, 4279–4302, 2017.

Yue, S., et al.: High Abundance of Fluorescent Biological Aerosol Particles in Winter in Beijing, China, ACS Earth Space Chem., 1, 493–502, 2017.

Yue, S., et al.: Abundance and diurnal trends of fluorescent bioaerosols in the troposphere over Mt. Tai, China, in spring, J. Geophys. Res.-Atmos., 124, 2018JD029486, 2019.

Regarding the factor analysis, it is not clear what the difference is between F1 and F5. Is F5 more primary anthropogenic emissions and F1 secondary anthropogenic emissions? The confusion is partly due to calling one anthropogenic and the other fossil fuel emissions – what is the difference between these two at this location.

Reply: We agree with you that F5 is more primary anthropogenic emissions and F1 mainly relate to secondary anthropogenic emissions. The differences between F1 and F5 can be explained from two aspects.

One is about the source profile. F5 is characterized by high values of OC, WSOC, Cl$^-$, SO$_4^{2-}$, NO$_3^-$, and NH$_4^+$. These components are mainly from direct emission and subsequent aging process of combustions of fossil fuel such as coal and oil gas. Different from F5, F1 is characterized by the highest level of DHOPA, which is the tracer for SOA from anthropogenic aromatics.

The other is about the source contribution. The relative contribution of F5 is much larger in winter than in summer, in accordance with the seasonal variations in primary emissions of fossil-fuel combustions. F5 contributed much more than F1 in winter and less than F1 in summer, indicating the stronger effect of secondary formation on F1. In addition, the contribution of F5 is more correlated with EC concentration ($r^2$=0.50), while F1 is less correlated with EC ($r^2$=0.29), again suggesting the important role of primary emission in F5 contribution.

In the revised manuscript, we change F5 from "fossil fuel combustion and aging processes" to "anthropogenic emission and aging" to avoid such confusion. The text has been revised as follows:

"The fifth factor (F5) had the largest abundances of OC, SO$_4^{2-}$, NO$_3^-$, NH$_4^+$, K$^+$, and Cl$^-$. These components are mainly from primary emission and subsequent aging processes of combustions of fossil fuel such as coal and oil gas. Therefore, F5 is likely to associate with the anthropogenic emission (i.e., fossil fuel combustion) and aging processes. The stronger effect of primary anthropogenic emissions on F5 can be supported by the significant correlation between F5 and EC concentration ($r^2$ = 0.50, $p$ < 0.01)."

The last line: The larger contributions of secondary BrC may also partly explain the lower MAE in summer (Table 1), since atmospheric aging processes would weaken light absorption (Zhong and Jang, 2014; Liu et al., 2016). In the summer there is also more sources of WSOC, such as biogenic SOA, that are not brown, which along with aging, would lower the MAE.

Reply: We agree with you that the larger contribution from biogenic SOA may result in lower MAE in summer. In the revised manuscript we have added the statements to explain the obvious differences in MAE in winter and summer. The revised texts are as follows (Line 224–227 and 502–505):

"The obvious seasonal variations in both $MAE_{365}$ and $k_{365}$ also suggest the distinct sources and formation mechanisms of BrC chromophores in different seasons. For example, in the summer aerosols, the larger contributions from biogenic SOA which are not or less light-absorbing will lower the MAE."

"Since fresh biogenic SOA are generally not or weakly light-absorbing, the larger contributions of biogenic SOA to WSOC would lower the MAE in summer (Table 1). In addition, atmospheric aging and photobleaching processes during the formation of secondary BrC would also result in the lower summer MAE (Zhong and Jang, 2014; Liu et al., 2016)."

---

## Author Response (AR2)

**Reply to Reviewer #2**

The authors have addressed most of my comments, except please make one more clarification before publication.

The text states: It is stressed that the results in the present work only represent the direct radiative effects of water-soluble fractions of BrC, while the estimated effects should be less than the actual effects of aerosol BrC since the water-insoluble species in BrC were not considered in the calculation.

Please add to this (if the authors agree) something like. In addition, the predicted radiative effects are for BrC measured in the extract and does not include changes in BrC light absorption if the BrC was in actual aerosol particles.

The point here is that I believe there are two issue that should be discussed, one that you only consider water-soluble species, and two, no attempt to convert bulk (extract solution) BrC to aerosol particle BrC. See Zeng et al for more details or references there-in.

**Reply:** We fully agree with you that light absorption properties of solvent extracted BrC and particle phase BrC are different. Previous studies have found the discrepancies between BrC light-absorption of particle- and extracted- BrC (Liu et al., 2013; Cheng et al., 2021; Zeng et al., 2022). Here, we add the clarification according to your suggestion in the revised manuscript. The corresponding references are also cited.

"It is stressed that the estimated effects in the present work only represent the direct radiative effects of water-soluble fractions of BrC, while the water-insoluble species in BrC were not considered in the calculation. Besides, these predicted radiative effects are for water-soluble BrC measured in the extract and do not represent the actual effects of atmospheric BrC aerosols. The discrepancies between BrC light-absorption of particle-phase BrC and extracted-BrC have been found in previous studies (Liu et al., 2013; Cheng et al., 2021; Zeng et al., 2022) and will not discussed here." (see Page 10, Lines 268-272)

References:
Liu, J., Bergin, M., Guo, H., King, L., Kotra, N., Edgerton, E., and Weber, R. J.: Size-resolved measurements of brown carbon in water and methanol extracts and estimates of their contribution to ambient fine-particle light absorption, Atmos. Chem. Phys., 13, 12389–12404, 2013.
Cheng, Z., Atwi, K., El Hajj, O., Ijeli, I., Al Fischer, D., Smith, G., and Saleh, R.: Discrepancies between brown carbon light-absorption properties retrieved from online and offline measurements, Aerosol Sci. Technol., 55, 92–103, 2021.
Zeng, L., Dibb, J., Scheuer, E., Katich, J. M., Schwarz, J. P., Bourgeois, I., Peischl, J., Ryerson, T., Warneke, C., Perring, A. E., Diskin, G. S., DiGangi, J. P., Nowak, J. B., Moore, R. H., Wiggins, E. B., Pagonis, D., Guo, H., Campuzano-Jost, P., Jimenez, J. L., Xu, L., and Weber, R. J.: Characteristics and Evolution of Brown Carbon in Western United States Wildfires, Atmos. Chem. Phys. Discuss. [preprint], in review, 2022.